# Mitochondria, Sex, and Cardiovascular Disease: A Complex Interplay

**DOI:** 10.3390/ijms26188971

**Published:** 2025-09-15

**Authors:** Andrea Iboleon-Jimenez, Alberto Contreras-Muñoz, Cristian Peláez-Berdún, Rafael Franco-Hita, Alba Sesmero, Ainhoa Robles-Mezcua, Jose M. García-Pinilla, Manuel Jimenez-Navarro, Mora Murri

**Affiliations:** 1Área de Gestión Sanitaria Este de Málaga-Axarquía, Vélez-Málaga, 29700 Malaga, Spain; andrea.iboleon@gmail.com; 2Faculty of Medicine, University of Malaga, 29010 Malaga, Spain; 3Cardiology and Cardiovascular Surgery Department, Virgen de la Victoria University Hospital, Campus de Teatinos s/n, 29010 Malaga, Spain; contrerasmu.alberto@gmail.com (A.C.-M.); cpelaez02@gmail.com (C.P.-B.); rafafranco9@hotmail.com (R.F.-H.); albasesmero98@icloud.com (A.S.); ainhoa.mezcua@gmail.com (A.R.-M.); mjimeneznavarro@gmail.com (M.J.-N.); 4Instituto de Investigación Biomédica de Málaga y Plataforma en Nanomedicina (IBIMA Plataforma BIONAND), 29590 Malaga, Spain; 5Centro de Investigación Biomédica en Red de Enfermedades Cardiovasculares (CIBERCV), Instituto de Salud Carlos III, 28029 Madrid, Spain; 6Unidad de Insuficiencia Cardíaca y Cardiopatías Familiares, Cardiología, Hospital Universitario Virgen de La Victoria, 29010 Malaga, Spain; 7Department of Dermatology and Medicine, Faculty of Medicine, University of Malaga, 29010 Malaga, Spain; 8Endocrinology and Nutrition UGC, Victoria Virgen University Hospital, 29010 Malaga, Spain; 9Centro de Investigación Biomédica en Red de la Fisiopatología de la Obesidad y Nutrición (CIBEROBN), Instituto de Salud Carlos III, 28029 Madrid, Spain

**Keywords:** cardiovascular diseases, mitochondria, mitochondrial dysfunction, sex, sex dimorphism

## Abstract

Cardiovascular diseases (CVDs) remain the leading cause of morbidity and mortality worldwide. Increasing evidence indicates that sex differences significantly influence the development, progression, and outcomes of CVDs. Recent advances have highlighted the central role of mitochondria, not only as cellular energy hubs but also as key regulators of oxidative stress, inflammation, and apoptosis, in mediating sex-specific cardiovascular responses. This review explores sexual dimorphism in cardiovascular disease, focusing on the interplay between mitochondrial function and sex hormones in cardiovascular tissues. We summarize current evidence on the molecular, hormonal, and cellular mechanisms contributing to sex-based disparities in cardiovascular outcomes. Preclinical studies suggest that female cardiac mitochondria may exhibit greater antioxidant capacity and produce fewer reactive oxygen species than male mitochondria, contributing to enhanced cardioprotection. Estrogen has been shown to influence mitochondrial bioenergetics and gene expression, affecting vascular tone, inflammation, and cardiac remodelling, whereas the role of testosterone remains less well defined. Additionally, sex-specific mitochondrial signalling responses have been reported under cardiac stress conditions, which may underlie differences in disease presentation and progression. A better understanding of how sex modulates mitochondrial function could improve risk stratification and support the development of personalized prevention and treatment strategies. Further research is needed to translate these mechanistic insights into clinical practice.

## 1. Introduction

### 1.1. General Background and Objective

Cardiovascular diseases (CVDs) remain the leading cause of global morbidity and mortality, accounting for an estimated 17.9 million deaths each year [1,2], with a disproportionate burden in low- and middle-income countries [3]. Despite recent advances in diagnostic and therapeutic strategies, the global prevalence of CVD continues to rise. This trend underscores the urgent need for novel preventive and therapeutic approaches.

CVDs comprise a broad spectrum of disorders affecting the heart and blood vessels, including coronary artery disease (CAD), cerebrovascular disease, peripheral arterial disease, rheumatic heart disease, and aortic atherosclerosis, among others. Although these conditions constitute a heterogeneous group of diseases, they often share common underlying mechanisms, particularly endothelial dysfunction and arteriosclerosis. Arteriosclerosis is characterized by the thickening, hardening, and loss of elasticity of arterial walls, often driven by age-related processes [4]. Atherosclerosis, a specific form of arteriosclerosis, involves a complex interplay of fibrous tissue, inflammatory cells, and lipid accumulation within the arterial intima, ultimately leading to the formation of atherosclerotic plaques [5]. Elevated plasma cholesterol, especially above 150 mg/dL, is a major driver of atherogenesis [6]. Both arteriosclerosis and atherosclerosis share common risk factors, including aging, hypertension, dyslipidemia, diabetes mellitus, smoking, physical inactivity, and obesity. Epidemiological studies indicate that the global incidence of CVD has increased over the past decade [7], in part due to inadequate management of these risk factors and the limited implementation of effective primary prevention strategies, especially in the early stages of life [8].

Mitochondrial dysfunction has emerged as a critical contributor to CVD pathogenesis by impairing ATP production, increasing reactive oxygen species (ROS) generation, and dysregulating calcium (Ca^2+^) homeostasis. Several studies have suggested that cardiac mitochondrial function is associated with cardiovascular disease, potentially exhibiting sex-specific differences [9,10]. Furthermore, several studies have highlighted the impact of estrogen deficiency on mitochondrial integrity and cardiac energy metabolism [11,12].

Despite growing awareness, outcome disparities in women with cardiovascular disease persist. Numerous recent studies show that women continue to receive less optimal treatment and experience poorer outcomes compared to men. As highlighted in The Lancet Commission’s 2021 report [13], CVDs in women remain understudied, under-recognized, underdiagnosed, and undertreated. In this review, we aim to integrate experimental and clinical evidence on the complex interplay between mitochondrial biology, sex hormones, and cardiovascular disease.

### 1.2. Method of Review

We conducted a literature search in PubMed and Web of Science to identify studies published until July 2025. Search terms included combinations of “cardiovascular disease”, “sex”, “sex hormones”, “testosterone”, “androgen”, “estrogen”, “mitochondria”, “mitochondrial dysfunction”, and “mtDNA”. Additional references were retrieved from the bibliographies of relevant articles. We prioritized peer-reviewed original research, though systematic reviews and meta-analyses were also included. Evidence was summarized narratively and organized thematically to integrate cardiovascular disease, sex, sex hormones, and mitochondrial biology, with a focus on mechanistic insights and translational implications.

## 2. Sex and Cardiovascular Disease

CVD, traditionally viewed as a male condition, is actually the leading cause of death and disability worldwide for both women and men. Despite this, sex and gender differences in CVD are frequently overlooked in clinical practice, contributing to its underdiagnosis, underrepresentation in research, and undertreatment in women [14]. As a result, women experience slower improvements in mortality rates compared to men, along with longer delays in emergency response, diagnosis, and revascularization procedures [15]. These disparities are attributed to both biological differences in symptom presentation and gender-related biases among patients and healthcare providers. Although awareness of the importance of sex and gender in health research is growing, substantial work remains to close these gaps.

The X chromosome plays a role in sex-specific CVD risk [16]. Analyses using UK Biobank data have revealed significant sex differences in genetic heritability for ~50% of binary traits linked to CVD. Therefore, this suggests that certain genetic variants may have a more pronounced effect on CVD risk in one sex than the other. These findings underscore the importance of incorporating sex-specific genetic markers to better understand disease susceptibility.

Sex differences in CVD can be intrinsic (innate) or acquired [16]. Intrinsic differences, present from birth, stem from biological sex and influence gene expression even before gonadal development, as shown in embryonic cardiomyocytes and endothelial cells from opposite-sex twins. These differences include sex-biased gene expression enriched in CAD-associated loci. In contrast, acquired differences emerge throughout life due to the effects of sex hormones or environmental influences. Studies in atherosclerotic tissues have identified sex-specific gene regulatory networks, with pro-atherogenic genes more active in immune cells in men and in endothelial and mesenchymal cells in women [17].

Owing to biological and hormonal factors, CVDs manifest differently in men and women. These sex-based differences influence the regulation of the cardiovascular system under both physiological and pathological conditions, contributing to variations in disease prevalence, clinical presentation, and outcomes [18]. These sex-based discrepancies are evident in epidemiological data (Table 1), which reveal substantial differences in clinical presentation and prognosis between men and women [19,20,21,22].

Several clinical studies [23,24,25,26,27] have advanced our understanding of the influence of biological sex and gender on the substrate of CVD. These studies highlight the importance of including sex as a key biological variable when assessing susceptibility, disease progression, and clinical outcomes. These studies have revealed significant sex-based differences in disease incidence and clinical manifestations, which are underpinned by structural, functional, and molecular distinctions within the cardiovascular system. Complementing these findings, experimental research using animal models [9,28,29,30] has provided mechanistic insight into how sex hormones contribute to cardiovascular dimorphism. This dimorphism arises from a complex interplay of hormonal, genetic, epigenetic, and environmental factors [25], and is reflected in distinct physiological and behavioral traits across sexes. The integration of both clinical and experimental data has been essential to uncover sex-specific mechanisms at the cellular and tissue levels, offering a valuable framework for interpreting sex-related differences in CVDs.

Although classical risk factors like increased lipid profiles, high blood pressure, and tobacco use are well-recognized contributors to CVD, accumulating research indicates that sex hormones are key regulators of cardiovascular function and disease development. Deciphering the dynamic relationship between these hormones and cardiovascular mechanisms is crucial to understanding the sex-specific differences in CVD onset, clinical features, and prognosis. Sex hormones play a critical role in cardiovascular physiology, including the regulation of blood pressure and cardiac function [31]. High levels of endogenous oestrogens in premenopausal women have been associated with a reduced risk of hypertension, obesity, diabetes, and vascular disease compared to men of similar age. However, this protective effect diminishes after menopause, reinforcing the role of sex hormones in cardiovascular health. In contrast, the impact of testosterone remains controversial. While some studies link testosterone therapy in older men to increased cardiovascular events [32], a meta-analysis that examined data from 106 studies, including over 15,000 participants, ensured that testosterone levels are not linked to an increase in major cardiovascular events [33]. Due to the scarcity of conclusive evidence, further well-designed, long-term studies are warranted to elucidate the impact of testosterone therapy on cardiovascular outcomes.

Interestingly, hormonal therapy in gender transition appears to influence CVD risk in transgender individuals [34]. Studies have shown that trans women undergoing long-term estrogen therapy may have an increased risk of ischemic stroke and, to a lesser extent, myocardial infarction, likely due to prolonged estrogen exposure [35]. Conversely, trans men did not show increased risk in some cohorts, although a Danish study reported higher CVD rates in both trans men and trans women compared to cisgender counterparts [36]. Notably, only about one-third of the increased CVD risk in trans men was attributable to hormone therapy, and gender-affirming hormone therapy was not a significant factor in trans women, possibly due to underreporting or low treatment prevalence. Additionally, a Dutch cohort revealed a twofold increase in mortality among both transgender women and men, with cardiovascular disease, particularly myocardial infarction, contributing to excess deaths in trans women [37].

## 3. Sex, Hormonal Regulation, and Mitochondrial Fitness

An increasing number of studies have demonstrated that mitochondrial function in the heart exhibits marked sexual dimorphism. Research using rodent models has revealed that although male cardiomyocytes often have a higher mitochondrial DNA (mtDNA) content and an elevated expression level of certain mitochondrial proteins, female mitochondria typically operate with greater intrinsic efficiency and generate lower levels of ROS under both basal and stress conditions [10,38]. For instance, experimental observations show that female ventricular mitochondria display lower mitochondrial Ca^2+^ uptake rates and enhanced assembly of respiratory chain supercomplexes, phenomena attributed in part to estrogen’s regulatory influence on several proteins [10,26,38]. Gene expression analyses further indicate that key genes involved in fatty acid oxidation, oxidative phosphorylation, and mitochondrial biogenesis are differentially regulated between the sexes, leading to unique baseline bioenergetic profiles and divergent responses to metabolic stress [22,39,40]. Consequently, the inherent sexual dimorphism in mitochondrial function not only influences normal myocardial energy metabolism but also predisposes males and females to distinct patterns of cardiovascular pathology [25,41].

Understanding mitochondrial processes and their sex-specific alterations is essential to gain deeper insight into the multifaceted roles of sex hormones within the cardiovascular system. Sex hormones, such as oestrogens and androgens, modulate multiple aspects of mitochondrial physiology [25]. Conversely, mitochondria are also essential for the production of steroid hormones, as they host the first step in sex hormone synthesis [42] (Figure 1). Sex hormones exert their effects through their respective receptors, estrogen receptors (ER) and androgen receptors (AR). Upon ligand binding, these nuclear receptors activate intracellular signalling cascades that ultimately regulate gene expression, thereby modulating a wide range of physiological processes [43]. These hormone-mediated mechanisms have been shown to exert specific effects on cardiac remodelling and the response to injury [24]. Estrogens are predominantly found in women, and testosterone in men. In both sexes, circulating levels of these hormones decline with age. Interestingly, aging is associated with a relative increase in estrogen levels in men and a predominance of testosterone over estrogen in postmenopausal women [44].

Estrogens exert pronounced cardioprotective effects by regulating mitochondrial biogenesis, function, and turnover [45,46,47]. They play both direct and indirect roles in modulating mitochondrial biogenesis via transcriptional coactivators such as peroxisome proliferator-activated receptor gamma coactivator 1-alpha (PGC-1α) and nuclear respiratory factors [45]. PGC-1α activates nuclear respiratory factor 1 and 2 (NRF1 and NRF-2), which in turn induce mitochondrial transcription factor A (TFAM), a key regulator of mtDNA replication and maintenance [46]. This signaling cascade enhances mitochondrial biogenesis and promotes the efficient assembly of electron transport chain (ETC) complexes [45]. Moreover, estrogen receptors, including ERα, ERβ, and the G protein-coupled estrogen receptor, are not only localized to the plasma membrane and nucleus but are also found within mitochondria, where they directly influence mitochondrial gene expression and function [26,39]. Specifically, estrogens, by enhancing PGC-1α, promote the transcriptional activity of estrogen-related receptor alpha (ERRα). ERRα, in turn, binds directly to the promoters of genes involved in fatty acid oxidation, the tricarboxylic acid (TCA) cycle, and the electron transport chain.

Beyond biogenesis, estrogens modulate critical mitochondrial functions, including ATP production, membrane potential, and calcium dynamics. The effects of estradiol are tissue-specific and depend on several factors, including the subtype of estrogen receptor (ERα versus ERβ), their subcellular localization, and the presence of selective estrogen receptor modulators. These modulators alter ER activity by inducing distinct receptor conformations and selectively recruiting coactivators or corepressors, thereby fine-tuning transcriptional programs that regulate mitochondrial function and cellular energy homeostasis.

Androgens also regulate mitochondrial function and biogenesis, although their role has received comparatively less attention than that of estrogens [48]. These hormones contribute to the regulation of glucose and lipid metabolism by modulating mitochondrial density, quality, and function. Interestingly, studies in hypogonadal men with diabetes, as well as in androgen-deficient male rodent models, have reported reduced mitochondrial density, structural abnormalities, and impaired ATP production [46]. Notably, exogenous testosterone administration has been shown to mitigate mitochondrial dysfunction and improve both glucose and lipid metabolic profiles. Testosterone has been reported to promote mitochondrial biogenesis in skeletal muscle through the activation of the androgen receptor (AR)/PGC-1α/TFAM pathway [49]. Preclinical studies show that castration reduces PGC-1α and TFAM protein levels, inhibiting mitochondrial biogenesis in rats, mice, and pigs. However, exogenous testosterone supplementation increases PGC-1α expression, enhances mitochondrial biogenesis, and restores mitochondrial function. In concordance, testosterone supplementation increases PGC-1α and TFAM levels in cultured C2C12 myogenic cells, which is attenuated by AR inhibitors [50]. Additionally, androgens and AR both influence mitochondria abundance by regulating mtDNA. Testosterone supplementation has been shown to restore mtDNA copy number, whereas androgen deficiency, as demonstrated in animal models of castration, results in a significant reduction in mtDNA copy number and impairs mitochondrial biogenesis [46]. These findings highlight the role of androgens in maintaining mitochondrial function and overall cellular energetics.

## 4. Mitochondrial Function in Cardiac and Vascular Tissues

In the healthy heart, cardiomyocytes rely on mitochondrial oxidative metabolism to meet the high energy demand required for contractile function. While developing cardiomyocytes primarily utilize glucose, adult cardiac cells predominantly depend on fatty acid oxidation to generate ATP [45]. Cardiac mitochondria are responsible for coupling the oxidation of metabolic substrates to ATP production via the ETC. This process of oxidative phosphorylation is remarkably efficient and critical for the heart, which requires a continuous energy supply to sustain contractile function. Beyond ATP synthesis, mitochondria play a pivotal role in intracellular Ca^2+^ homeostasis by rapidly buffering cytosolic Ca^2+^ through uptake via the mitochondrial Ca^2+^ uniporter, and this response is particularly important during β-adrenergic stimulation when Ca^2+^ fluctuations occur [38,51]. The maintenance of mitochondrial membrane potential and the proper assembly of respiratory supercomplexes are essential for minimizing electron leakage, thereby reducing the production of ROS under both basal and stress conditions. These properties are fundamental for ensuring effective cardiac contractility and preserving overall cardiovascular homeostasis [52,53]. Furthermore, mitochondrial quality control mechanisms, such as biogenesis, fission/fusion dynamics, and mitophagy, sustain optimal mitochondrial performance and allow rapid adaptation to changes in energy demand [54].

Mitochondrial dysfunction has emerged as a critical contributor to cardiovascular disease by impairing energy production, promoting apoptotic signalling, reducing mtDNA content, and exacerbating oxidative stress, all of which negatively affect cell survival and function [45,55]. In fact, disruptions in metabolic flexibility and substrate preference are increasingly recognized as early events in the pathogenesis of CVD [45,46]. In pathological states, defects in the ETC can lead to decreased ATP synthesis, increased electron leakage, and subsequent overproduction of ROS, which promotes damage to mitochondrial membranes, proteins, and nucleic acids, thereby compromising mitochondrial integrity [25,56]. This increased ROS generation also triggers the opening of the mitochondrial permeability transition pore, which facilitates the release of pro-apoptotic factors such as cytochrome c and activates caspase-dependent cell death pathways [56,57]. Reductions in mitochondrial mass coupled with impaired activities of key ETC complexes have been associated with diminished myocardial contractility and adverse ventricular remodeling in the failing heart [58,59]. Aging further exacerbates mitochondrial dysfunction through accumulation of deleterious mtDNA mutations and impaired turnover, leading to progressive bioenergetic decline, an effect that is particularly pronounced in postmenopausal women due to estrogen loss [9,60].

Mitochondria constantly undergo fission and fusion in response to environmental signals, and these dynamic processes are essential for maintaining mitochondrial function and overall cellular health. Specifically, hormones such as estrogen and testosterone influence mitochondrial biogenesis and dynamics, which are essential for maintaining cellular function and survival under both normal and stress conditions [45]. These regulatory effects are particularly important in the heart, where alterations in mitochondrial function, such as impaired fusion and excessive fission, are linked to various forms of cardiac injury and dysfunction, including ischemia/reperfusion injury, cardiomyopathy, and metabolic disturbances. Maintaining proper mitochondrial dynamics is crucial for preserving cardiac health and mitigating the progression of CVD [45]. Moreover, endothelial dysfunction, a key factor in atherosclerosis development, is characterized by increased oxidant production, like ROS, which leads to lipid peroxidation, mtDNA damage, and altered mitochondrial dynamics. Damaged mtDNA can regulate mitochondrial dynamics by either inducing or inhibiting mitochondrial dynamics-related proteins, processes that are early events in atherosclerosis and may contribute to disease progression. Furthermore, changes in the gene expression of mitochondrial dynamics-related genes (Drp1, Mfn1, Mfn2, and Opa1) influence the proliferation and migration of vascular smooth muscle cells, key factors in pathological hypertrophy of the arterial wall. These findings highlight that both mitochondrial fission induction and fusion inhibition are hallmark features of CVD, including heart failure and atherosclerosis.

## 5. Cardiovascular Disease, Mitochondria, and Sex-Specific Factors

Sex hormones have been shown to regulate mitochondrial dynamics, metabolism, and inter-organelle communication processes [45] that are increasingly recognized as key contributors to CVD (Figure 2). Androgens promote mitochondrial function to maintain energy balance in males, while estrogens exert similar effects in females through mitochondrial regulation [61,62,63,64]. Both estrogen in females and testosterone in males have been shown to upregulate antioxidant enzymes, counteracting oxidants, and suggesting a cardioprotective role mediated by sex hormones [45,65]. Testosterone deficiency negatively affects interfibrillar mitochondrial performance in cardiac tissue, weakens myocardial contractility, and increases oxidative stress [66]. Nonetheless, additional studies are needed to better delineate the specific regulatory roles of sex hormones, particularly testosterone, in mitochondrial function and to understand their contribution to sex-related differences in CVD.

Clinical and interventional studies have provided strong evidence linking mitochondrial dysfunction to cardiovascular disease [18,25]. Beyond their role as the primary source of cellular energy, mitochondria are involved in numerous cellular processes. Alterations in mitochondrial bioenergetics and oxidative stress have been associated with sex-related differences in the prevalence, progression, and treatment response of various pathologies, such as cardiovascular diseases. For example, the cardioprotective effects of vitamin E are age- and sex-dependent, potentially mediated by differential regulation of mitochondrial pathways involved in oxidative stress and apoptosis following ischemia/reperfusion injury [67]. Moreover, sex-specific differences in mitochondrial function, including reactive oxygen species production, membrane potential, and mitophagy, may underlie divergent cardiovascular responses to stress and injury in males and females [65]. In parallel, studies employing non-invasive imaging techniques and biomarker assessments have identified altered expression of mitochondrial proteins and reduced ETC activity as early indicators of myocardial dysfunction, suggesting that mitochondrial-derived biomarkers may serve as valuable tools for early diagnosis and risk stratification in CVD [68,69]. Experimental rodent models have further corroborated these observations; for instance, genetically diverse mouse populations have revealed strong correlations between the expression of mitochondrial genes and diastolic function traits, with females displaying a distinctive pattern of mitochondrial performance that predisposes them to specific forms of heart failure under stress [10,23]. Moreover, studies in female rats show reduced ROS production and enhanced antioxidant capacity in cardiac mitochondria compared to males, leading to greater cardioprotection [29]. Both animal and human studies [28,70,71] suggest that sexual dimorphism in cardiovascular disease is mediated, in part, by differences in mitochondrial structure, function, and redox regulation.

## 6. Therapeutic Strategies Targeting Mitochondrial Dysfunction

Recent insights into the role of mitochondrial dysfunction in cardiovascular disease have led to increasing interest in therapeutic approaches designed to reestablish mitochondrial homeostasis and improve cardiac outcomes. As previously described, one promising strategy involves the use of mitochondria-targeted antioxidants, such as MitoQ, which have been shown to attenuate ROS production, improve mitochondrial respiration, and alleviate diastolic dysfunction in both preclinical and clinical settings [72,73]. Additionally, pharmacologic agents that stimulate mitochondrial biogenesis also represent a key therapeutic avenue. Compounds that stimulate transcriptional coactivators such as PGC-1α and transcription factors like TFAM have the potential to enhance oxidative phosphorylation and reduce intracellular oxidative stress [74,75]. Hormone replacement therapies, particularly estrogen supplementation in postmenopausal women, have been evaluated for their capacity to preserve mitochondrial function. While estrogen has demonstrated beneficial effects on mitochondrial biogenesis and antioxidant systems, clinical outcomes remain inconsistent, highlighting the complex regulation of mitochondrial dynamics by sex hormones [26,76]. In addition, small-molecule modulators of mitochondrial dynamics, designed to favor fusion processes and limit excessive fission, are being investigated for their ability to restore mitochondrial network integrity and support bioenergetic function in dysfunctional cardiac tissue [45,77]. Parallel to pharmacological strategies, lifestyle interventions such as aerobic exercise and dietary modification have been shown to stimulate mitochondrial biogenesis and enhance oxidative metabolism, mitigating cardiovascular risk. Importantly, these effects appear to be modulated by the hormonal milieu, with sex-specific differences reported in mitochondrial adaptations [22,78]. Regular aerobic exercise, for example, has been shown to stimulate mitochondrial biogenesis, enhance oxidative capacity, and improve overall energy metabolism in both skeletal muscle and cardiac tissue [79,80]. Importantly, studies indicate that these adaptations may differ between the sexes, with female subjects often exhibiting enhanced mitochondrial responses and substrate flexibility in response to exercise training, a benefit that is mediated, at least in part, by estrogen signaling [81,82]. Collectively, these emerging strategies underscore the importance of targeting mitochondrial dysfunction through both pharmacological and non-pharmacological means, while considering sex-based differences to optimize cardiovascular therapeutic outcomes.

## 7. Integrating Evidence of Sex-Specific Treatment Responses

The impact of sex on therapeutic response in CVD remains an area of ongoing debate. First, several high-quality studies support the comparable efficacy of standard treatments across sexes. The Cholesterol Treatment Trialists’ (CTT) Collaboration demonstrated equivalent reductions in cardiovascular risk with statin therapy in men and women [83], a finding corroborated by subsequent large-scale meta-analyses [84,85]. Statins, although they appear to increase HDL-C levels more in women than in men, demonstrate comparable effects on other lipid parameters across sexes [86]. Similarly, the TALOS-AMI trial [87] reported no significant sex-based differences in either efficacy or safety following de-escalation of dual antiplatelet therapy (DAPT) after acute myocardial infarction. However, this interpretation should be made with caution, given the unequal sex representation in the study population (217 women vs. >1110 men). A sex-stratified meta-analysis of shortened versus prolonged DAPT durations following drug-eluting stent implantation likewise confirmed consistent outcomes across sexes [88]. Another meta-analysis conducted by Piccolo et al. [89] reported that the efficacy and safety profiles of antithrombotic therapies were consistent in both female and male patients, suggesting the absence of clinically relevant sex-related heterogeneity. In PRODIGY, a randomized clinical trial of the efficacy and safety of antithrombotic therapies in patients with or without peripheral arterial disease, sex did not modify the treatment effect, showing similar ischemic and bleeding outcomes in men and women between long (24 months) and short (6 months) duration of dual antiplatelet therapy after percutaneous coronary intervention (PCI) [90], suggesting no clinically relevant sex-related heterogeneity. Furthermore, detailed analysis of patients according to baseline troponin concentration by Boersma et al. [91] showed no sex-related differences in treatment response, with glycoprotein IIb/IIIa inhibitors reducing the 30-day risk of death or myocardial infarction equally in men and women with elevated troponins. Other observational studies in hypertension and heart failure have similarly reported largely comparable treatment responses or inconclusive differences between sexes [92].

Second, other studies highlight potential sex-specific variations in treatment response and safety profiles. These differences can be observed from the medical care provided to women with STEMI (longer care and treatment times) to 30-day mortality, especially in STEMI. Female sex was an independent predictor of overall mortality, even after adjustment for other factors [93]. Women have been reported to experience higher bleeding risks under DAPT [94] and may present with worse short-term procedural outcomes following PCI, although long-term survival benefits appear comparable [95]. These differences, although sometimes modest, underscore the importance of careful consideration of sex-specific clinical trajectories.

Taken together, these apparently conflicting results emphasize both the robustness of current guideline-based treatments and the need for a nuanced interpretation of sex-related effects. The underrepresentation of women in cardiovascular clinical trials, heterogeneity in study designs, and differing endpoints may partly explain inconsistencies across studies. Nonetheless, the emerging consensus is that while guideline-directed therapies benefit both sexes, tailoring interventions based on sex-specific biology and comorbidities may enhance outcomes.

## 8. Challenges and Opportunities in Translational Research

Despite significant advances in understanding the molecular underpinnings of sex-dependent mitochondrial function, several challenges remain in translating these insights into clinical practice. The inherent complexity of mitochondrial regulatory networks, which span transcriptional control, dynamic structural rearrangements, and post-translational modifications, poses considerable difficulties in defining robust biomarkers or therapeutic targets. Moreover, the majority of preclinical studies have historically been conducted in predominantly male animal models, highlighting the need for sex-balanced experimental designs that accurately reflect the human condition. Further complicating translational efforts is the multifactorial nature of cardiovascular disease, which involves interactions between mitochondrial dysfunction, systemic inflammation, metabolic derangements, and neurohormonal alterations. Accordingly, future research must adopt multidisciplinary approaches that integrate high-throughput omics analyses, advanced imaging techniques, and precise functional assays to dissect the contributions of mitochondria under diverse pathological states. Such efforts will be critical not only for identifying novel molecular targets but also for the development of personalized therapeutic strategies that incorporate sex-specific considerations.

## 9. Implications for Precision Cardiovascular Medicine

The recognition of sexual dimorphism in mitochondrial biology has profound implications in cardiovascular care, opening new avenues for precision medicine. Personalized therapeutic interventions that consider sex, hormonal status, and mitochondrial phenotype may enhance treatment responses and minimize adverse outcomes in CVD. For instance, tailored administration of mitochondrial-targeted antioxidants or biogenesis enhancers may provide superior cardioprotection when optimized for the sex-specific differences observed in mitochondrial ROS production and respiratory efficiency. Similarly, hormone replacement strategies aimed at restoring estrogen levels in postmenopausal women could potentially reverse mitochondrial deterioration, although careful evaluation of downstream effects is warranted. Advances in non-invasive diagnostic techniques that facilitate real-time assessments of mitochondrial function, such as metabolic imaging and circulating biomarker assays, may further enable early detection of mitochondrial dysfunction, allowing clinicians to implement interventions before irreversible cardiac damage occurs. Ultimately, the integration of mitochondrial biology into the broader framework of personalized cardiovascular medicine represents a promising frontier that may help reduce sex-based disparities in CVD risk and improve clinical outcomes for both men and women.

## 10. Future Directions in Translational Cardiovascular Research

Several research directions merit exploration to fully elucidate the complex interplay among mitochondria, sex, and cardiovascular disease. First, large-scale studies specifically designed to capture sex-specific differences in cardiac mitochondrial function are essential. These studies should involve diverse patient cohorts and leverage next-generation sequencing, mass spectrometry, and high-resolution imaging methodologies to comprehensively characterize mitochondrial dynamics under normal and pathological states. Second, preclinical models that incorporate both sexes with balanced representation and that simulate the human hormonal environment will be invaluable for uncovering new molecular regulators of mitochondrial biogenesis, dynamics, and oxidative stress responses. Such models may also aid in the discovery of novel therapeutic targets and in the validation of candidate drugs designed to restore mitochondrial function. Third, clinical trials evaluating mitochondrial-targeted interventions should stratify patient outcomes by sex, ensuring that the observed effects of treatments such as MitoQ supplementation or hormone replacement therapy are appropriately contextualized within a sex-specific framework. Fourth, the multifactorial nature of cardiovascular disease complicates translational efforts. Addressing this complexity will require interdisciplinary research that bridges molecular biology, endocrinology, and clinical cardiology, which will be vital for translating laboratory findings into actionable therapies that improve patient care and reduce health disparities. Fifth, current therapeutic strategies, supported by strong evidence, are equally recommended for both sexes, but growing recognition of subtle differences in efficacy and safety highlights an opportunity for greater personalization. Future clinical trials must ensure balanced sex representation and include pre-specified, adequately powered sex-stratified analyses. Finally, efforts to integrate mitochondrial biomarkers into routine diagnostic protocols may allow early identification of patients at risk for CVD, enabling timely and targeted interventions based on an individual’s unique mitochondrial and hormonal profile. Combining these advances with insights from sex-specific genetics will be key to advancing precision medicine in cardiology.

## 11. Conclusions

In summary, CVD remains a leading cause of morbidity and mortality worldwide, exhibiting notable sex-based differences in prevalence, clinical presentation, and outcomes. Mitochondrial dysfunction has emerged as a central pathogenic mechanism underlying CVD, contributing to impaired bioenergetics, increased oxidative stress, inflammation, and cell death. The interplay between sex hormones and mitochondrial biology plays a pivotal role in shaping these sex-specific disparities, influencing disease onset, progression, and response to therapy. At the same time, current evidence indicates that established therapies benefit both sexes. Nevertheless, integrating sex-specific biological and metabolic factors into clinical decision-making may further optimize efficacy, safety, and equity in cardiovascular care. Despite significant advances, further research is needed to elucidate sex-specific mitochondrial mechanisms in CVD and to develop targeted interventions. Future efforts should emphasize sex-stratified basic, translational, and clinical studies to better characterize mitochondrial function and dysfunction across sexes. Integrating these insights into clinical practice holds great promise for advancing personalized, sex-informed therapeutic strategies that improve prevention, treatment, and ultimately, cardiovascular outcomes for both women and men.

## Figures and Tables

**Figure 1 ijms-26-08971-f001:**
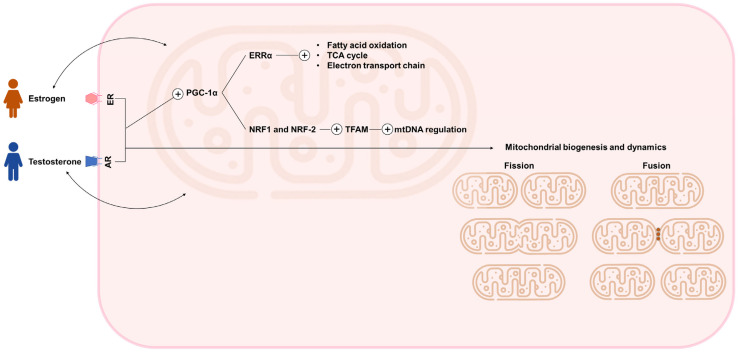
Sex hormone-dependent regulation of mitochondrial biogenesis, metabolism, and dynamics. AR, androgen receptor; ER, estrogen receptor; ERRα, estrogen-related receptor alpha; mtDNA, mitochondrial DNA; NRF1/2, nuclear respiratory factor 1 and 2; PGC-1α, peroxisome proliferator–activated receptor gamma coactivator 1-alpha; TFAM, mitochondrial transcription factor A; TCA, tricarboxylic acid. The figure was designed using PowerPoint. Icons representing a woman, a man, and mitochondria were adapted from Freepik (www.flaticon.com) (accessed on 28 May 2025).

**Figure 2 ijms-26-08971-f002:**
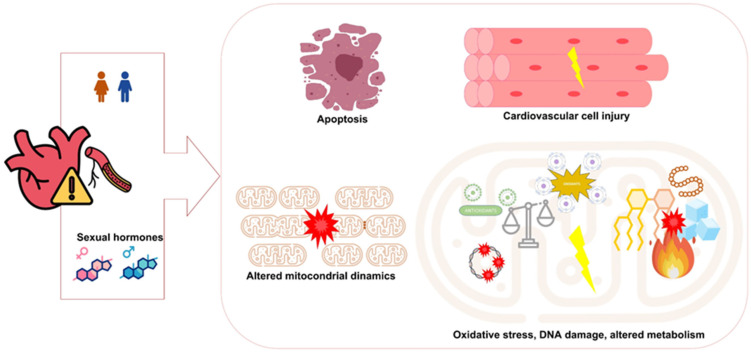
Interplay between sex hormones, mitochondrial function, and cardiovascular disease. The figure was designed using PowerPoint. Some of the icons were adapted from Freepik, Ranksol, and clinihcDinosoftLabs graphics (www.flaticon.com) (accessed on 28 May 2025).

**Table 1 ijms-26-08971-t001:** Comparison of cardiovascular disease between men and women.

Condition	Men	Women
**Overall CVD prevalence**	Higher	Lower
**Age of CVD onset**	Earlier onset	Later onset, particularly after menopause
**Heart failure**	Lower prevalence Predominantly HFrEF	Higher prevalence Predominantly HFpEF
**Myocardial infarction**	Higher incidence and mortality Typical chest pain presentation.	Lower incidence but higher post-MI mortality Atypical symptoms (fatigue, nausea, jaw pain) Increased risk of delayed diagnosis.
**Valvular heart disease** **Mitral valve** **Aortic valve**	Lower prevalence Higher risk of progression Higher prevalence Earlier presentation More likely to receive invasive treatment Better prognosis	Higher prevalence Lower risk of progression Lower prevalence Later presentation More conservative treatment approach Worse prognosis
**Peripheral artery disease**	Higher prevalence Earlier onset Typical symptoms (intermittent claudication) Adequate treatment Better prognosis	Lower prevalence (potential underdiagnosis) Later onset Atypical symptoms Less likely to receive appropriate treatment Worse prognosis
**Stroke**	Higher incidence at younger ages Better prognosis	Higher incidence at older ages Worse prognosis

CVD, cardiovascular disease; HFpEF, heart failure with preserved ejection fraction; HFrEF, heart failure with reduced ejection fraction.

## Data Availability

The data underlying this article will be shared on reasonable request by the corresponding author.

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
