# Peer review of "Mitochondria, Sex, and Cardiovascular Disease: A Complex Interplay"

_ijms, 2025, doi:10.3390/ijms26188971_

Round 1

Reviewer 1 Report

Comments and Suggestions for Authors

The current manuscript discusses how biological sex intersects with mitochondrial biology to shape cardiovascular disease (CVD) risk, presentation, and outcomes. It synthesizes evidence across epidemiology, genetics, hormone signaling, and organelle physiology; highlights estrogen/androgen effects on mitochondrial biogenesis, ROS handling, and energetics; and discusses sex-aware therapeutic avenues, the translational hurdles, and implications for precision cardiology. It addresses the major topic about why sex matters in CVD, provides mitochondrial mechanisms in a clear and concise way, derives the bottom-line conclusion as sex-informed mitochondrial mechanisms are central to CVD and should guide prevention and therapy. It reads, however, more into a narrative synthesis without an explicit, reproducible search strategy or inclusion criteria, the paper does not conclusively map the literature landscape, which limits how definitively it “closes” the gap. Authors should consider adding a transparent methods part, including databases, dates, search terms, screening approach, and how evidence was prioritized/summarized. Also, minor formatting/consistency problems in the citation list and occasional style/grammar slips occur, which should be corrected in the following edited version.

Author Response

Query 1. The current manuscript discusses how biological sex intersects with mitochondrial biology to shape cardiovascular disease (CVD) risk, presentation, and outcomes. It synthesizes evidence across epidemiology, genetics, hormone signaling, and organelle physiology; highlights estrogen/androgen effects on mitochondrial biogenesis, ROS handling, and energetics; and discusses sex-aware therapeutic avenues, the translational hurdles, and implications for precision cardiology. It addresses the major topic about why sex matters in CVD, provides mitochondrial mechanisms in a clear and concise way, derives the bottom-line conclusion as sex-informed mitochondrial mechanisms are central to CVD and should guide prevention and therapy. It reads, however, more into a narrative synthesis without an explicit, reproducible search strategy or inclusion criteria, the paper does not conclusively map the literature landscape, which limits how definitively it “closes” the gap. Authors should consider adding a transparent methods part, including databases, dates, search terms, screening approach, and how evidence was prioritized/summarized. Also, minor formatting/consistency problems in the citation list and occasional style/grammar slips occur, which should be corrected in the following edited version.

Answer. We sincerely thank the reviewer for the positive assessment of our manuscript and for the constructive suggestions. We apologize for the missing methodological details and formatting issues, and we have carefully addressed each point as follows. A dedicated Method of review section has now been included (Marked manuscript, Page 4, lines 37–42 and Page 5, lines 44–45), describing the databases consulted, the search terms used, the period covered, and the criteria for prioritizing evidence. The reference list has been thoroughly revised to ensure uniform formatting according to the journal’s guidelines, and all in-text citations have been checked for consistency. Furthermore, the manuscript has undergone careful language editing to correct minor grammatical issues and enhance overall readability.

Reviewer 2 Report

Comments and Suggestions for Authors

The paper is well written, the authors did a great job on the reference search and the construction of the manuscript. 

However, major revisions are needed:

  • Table 1, described on page 3, line 113, is not present in the document.
  • Figures require a legend and information on whether they were created using a scientific drawing app or database.
  • Page 5, line 183, reference 24 is not in the correct format
  • It would be beneficial to include studies that did not observe a sex difference in CVD treatment as part of the discussion.  I felt that a discussion of studies that either did not find a change between sexes in CVD treatment or studies that were inconclusive could have been better addressed, particularly concerning gender considerations. Worth adding some studies that did not see a sex difference for CVD treatment as a point of discussion.

Author Response

The paper is well written, the authors did a great job on the reference search and the construction of the manuscript. 

Answer. We sincerely thank the reviewer for the encouraging comments and for the valuable suggestions, which have substantially improved the manuscript. All the requested changes have been carefully addressed.

However, major revisions are needed:

Query 1. Table 1, described on page 3, line 113, is not present in the document.

Answer. We thank the reviewer for the comment and we apologize for the confusion. Table 1 is already included in the manuscript (marked manuscript, page 31), where it summarizes the epidemiological data referred to in the text. We have checked the revised version to ensure that the table is clearly visible.

Query 2. Figures require a legend and information on whether they were created using a scientific drawing app or database.

Answer. We thank the reviewer for this important suggestion. We have revised the figure legends accordingly.

Query 3. Page 5, line 183, reference 24 is not in the correct format.

Answer. We thank the reviewer and apologize for this formatting error. Reference 24 has now been corrected (marked manuscript, page 9, line 145).

Query 4. It would be beneficial to include studies that did not observe a sex difference in CVD treatment as part of the discussion. I felt that a discussion of studies that either did not find a change between sexes in CVD treatment or studies that were inconclusive could have been better addressed, particularly concerning gender considerations. Worth adding some studies that did not see a sex difference for CVD treatment as a point of discussion.

Answer. We thank the reviewer for this insightful suggestion. We have incorporated additional discussion highlighting studies that did not observe sex differences in CVD treatment, as well as inconclusive findings. These additions provide a more balanced perspective on gender considerations and are reflected in the revised manuscript (marked manuscript, pages 17-19, lines 312–354) Additionally, the Future Directions… and Conclusions sections have also been updated to emphasize the importance of integrating sex-specific factors and ensuring adequate representation of both sexes in future studies (marked manuscript, page 21, lines 407–411 and 424–426).

Round 2

Reviewer 2 Report

Comments and Suggestions for Authors

The authors were able to address all previous questions, and I have no further questions about the manuscript.